# Essential Oil Composition and Antioxidant Activity of Oregano and Marjoram as Affected by Different Light-Emitting Diodes

**DOI:** 10.3390/molecules28093714

**Published:** 2023-04-25

**Authors:** Danial Sarfaraz, Mehdi Rahimmalek, Mohammad R. Sabzalian, Shima Gharibi, Adam Matkowski, Antoni Szumny

**Affiliations:** 1Department of Agronomy and Plant Breeding, College of Agriculture, Isfahan University of Technology, Isfahan 84156-83111, Iransabzalian@iut.ac.ir (M.R.S.); 2Department of Horticulture, College of Agriculture, Isfahan University of Technology, Isfahan 84156-83111, Iran; 3Department of Food Chemistry and Biocatalysis, Wrocław University of Environmental and Life Sciences, 50-375 Wrocław, Poland; 4Core Research Facilities (CRF), Isfahan University of Medical Sciences, Isfahan 81746-73461, Iran; 5Department of Pharmaceutical Biology and Biotechnology, Botanical Garden of Medicinal Plants, Wroclaw Medical University, 50-367 Wrocław, Poland

**Keywords:** oregano, marjoram, essential oil, phenolics, illumination spectrum, GC-MS, antioxidant

## Abstract

Oregano and marjoram are important aromatic spices in the food industry, as well as medicinal plants with remarkable antioxidant properties. Despite their popularity, little is known about treatments that would influence the antioxidant capacity of essential oils. In this study, different spectra of LED light, namely blue, red, white, blue-red, and natural ambient light as a control, were applied to assess the essential oil content, composition, flavonoid, phenolic, and antioxidant capacity of oregano and marjoram. GC-MS analysis revealed thymol, terpinen-4-ol, sabinene, linalool, p-cymene, and γ-terpinene as the main compounds. In oregano, the thymol content ranged from 11.91% to 48.26%, while in marjoram it varied from 17.47% to 35.06% in different samples. In oregano and marjoram, the highest phenolic contents were in blue (61.26 mg of tannic acid E/g of DW) and in white (65.18 mg of TAE/g of DW) light, respectively, while blue-red illumination caused the highest increase in total flavonoids. The antioxidant activity of oregano and marjoram extract was evaluated using two food model systems, including DPPH and β-carotene bleaching. The highest antioxidant capacity was obtained in control light in oregano and blue-red light in marjoram. The results provide information on how to improve the desired essential oil profile and antioxidant capacity of extracts for industrial producers.

## 1. Introduction

Essential oils (EOs) are the major components of many medicinal and aromatic plants of Lamiaceae. In addition to these main substances, aromatic spices and medicinal plants, such as oregano and marjoram, contain significant amounts of phenolic antioxidants, such as hydroxycinnamic and flavonoid compounds. Thymol and carvacrol are two valuable aromatic monoterpenes with different applications in the food and pharmaceutical industries [1]. These components have been found mainly in some medicinal and spice plants, such as thyme [2], oregano [3], and ajowan [4] seeds. Today, the production of a large quantity of essential oils as well as phenolic compounds under control conditions is of great interest in the context of their use as spices, food antioxidants, and biopreservatives, as well as food supplements and pharmaceuticals for preventing diseases associated with oxidative stress.

Marjoram (*Origanum majorana*) originates from the eastern Mediterranean, whereas Oregano (*O. vulgare* L.) occurs naturally in most of Eurasia. Both are perennial species of Lamiaceae, cultivated mostly in Mediterranean regions but also further north. These aromatic plants have been used as spices for special flavoring in foods. The major components of these plants include thymol, carvacrol, and methyleugenol [5]. Different applications of these components as spices in food technology have been reported for their desirable aroma [6]. Besides food applications, antioxidant, antifungal, antibacterial, and anti-inflammatory activities of these components that are valuable in the production of drugs have been widely mentioned in the literature [7]. Therefore, increasing and improving these compounds in oregano has been considered a major goal for many industrial producers.

Light is the essential factor that controls the growth, development, and metabolism of plants, which are affected by irradiance at all stages of growth [8]. Variable light conditions are important in various latitudes and are related to global climate changes as well. Various wavelengths can influence the metabolism in a complex way and excess irradiation by certain spectra can trigger detrimental oxidative processes in the organism. Therefore, understanding the relationships between the light spectrum and the chemical composition of aromatic constituents in industrially valuable plants is important. In this regard, the use of different light-emitting diode (LED) systems provided new insights into the control of metabolic profiles in medicinal and aromatic plants [9]. LEDs with different wavelengths and energy density (flounce) of light can provide various environmental conditions that influence plant metabolism. Some LED frequencies can be used as signaling factors to increase the essential oils and polyphenolic compounds content [10]. The use of LEDs to improve phytochemical composition has been reported in many medicinal and aromatic plants, including *Thymus vulgaris* [11], *Melissa officinalis* [10], *Ocimum basilicum* [12], *Perovskia abrotanoides* [13], and *Mentha longifolia* [14]. However, there are no reports on the effects of LED illumination on phytochemical composition and antioxidant activity in oregano and marjoram.

The goals of this research were to assess the influence of four light spectra: red, blue, blue-red, and white on essential oil yield, chemical composition, phenolic, and flavonoid content, and the antioxidant capacity of these two species based on two antioxidant model systems.

## 2. Results and Discussion

### 2.1. Essential Oil Content

Analysis of variance revealed significant differences for oil, phenolic, flavonoid, and antioxidant capacity (Table 1). The oil content showed a significant variation as a result of the LED treatments. The highest and lowest oil content were obtained in red (OV = 2.89) and white (OM = 0.94) light, respectively. However, the two studied species revealed different responses to light treatments. In O. vulgare, the highest oil content was observed in red light, while the lowest was obtained in blue light (Figure 1). In O. majorana, the blue-red treatment led to an elevated oil content compared to others (Figure 1).

Both species revealed different trends in terms of oil content in different spectra (Figure 1). In the previous research, red and blue-red illumination resulted in a higher increase in oil content than the other light spectra such as *Thymus* in red light [11], *Mentha longifolia* [14], and *Ocimum gratissimum* [15]. However, in another species of Lamiaceae, *Perovskia abrotanoides*, the essential oil content increased in response to white light [13].

The probable mechanism for red light is that it can induce oil biosynthesis through exogenous primary precursors through stimulation. Another possible interpretation is that the wavelength of light might lead to an increase in oil content as a stress condition [11]. In Lamiaceae, secretory trichomes are the main source of essential oil, thus it can be suggested that light can affect trichome morphology and distribution. Similar effects were also observed in thyme [11] and *Ocimum* [15]. Secretory tissues in the Lamiaceae family are considered the most important tissue for the accumulation of essential oils. The glandular trichomes are determined as the site of oil production. High energy lights can destroy this trichome structure and lead to the release of oils [16]. However, trichome morphology and distribution are considered crucial factors affecting oil production [17]. Finally, most plants can display an acclimation strategy, such as altering their essential oil content to cope with different lighting conditions as the mechanisms of stress response [10].

### 2.2. Essential Oil Composition

High variation was observed between treatments. The main oil components in the two studied species were thymol, terpinen-4-ol, sabinene, linalool, *p*-cymene, and γ-terpinene (Table 2). In oregano, the thymol content ranged from 11.91% in control to 48.26% in blue light, while in marjoram it varied from 17.47% in red to 35.06% in blue-red light. In a similar experiment in four Thymus species, Tohidi et al. [11] revealed that blue light was more effective in increasing the thymol content, similar to the values obtained for oregano. A similar trend was also observed for p-cymene (Table 2). However, these trends could be variable in different species. In the present investigation, thymol and linalool were decreased in marjoram treated with red light (Table 2), while Nguyen and Saleh [18] also revealed an increase in α-pinene, limonene, and carvone in mint under red light. Furthermore, in oregano, blue light led to a decrease in p-cymene and an increase in thymol content. Previous literature also confirmed that blue LED lights can cause oxidative damage in some compounds and their conversion [18]. For example, in oregano, thymol production is promoted by increased consumption of *p*-cymene as a substrate (Table 2).

The composition can be affected by many environmental factors, including the period of light, the intensity, and the range in the light spectrum [10]. Most of the detected compounds are monoterpenes. A probable interpretation for the increase in monoterpenes as secondary metabolites is a stress condition causing the plant to produce a greater number of compounds. A lower wavelength spectrum, such as blue light, has higher energy, and consequently, high energy can act as stress for the plant [10]. According to the energy dissipation mechanism suggested by Selmar and Kirinwächter [19] and described by Gharibi et al. [20], the exceeded energy of CO_2_ fixation can be applied for the production of specific, nonprimary metabolites, and the high energy wavelength can lead to increased monoterpene production through the mevalonic acid (MVA) and methylerythritol phosphate (MEP) pathways [10].

Moreover, the production as well as isomer conformation can perform a crucial role in the decrease or increase in the production of a final product through light treatments [13]. For example, Ahmadi et al. [10] showed blue-red light as the main treatment to increase the monoterpenes. However, some researchers have reported that chemotypes and plant species are more important factors for oil component variation [21].

### 2.3. Total Phenolic and Flavonoid Content

A high variation was obtained between treatments in both species. In oregano, the highest and lowest TPC was under blue (61.26 mg of tannic acid E/g DW) and white (21.74 mg of TAE/g DW) light, respectively. In marjoram, it ranged from 20.50 in blue-red to 65.18 (mg TAE/g DW) in white light. The highest and lowest flavonoid content (TFC) was obtained in blue-red and white light, respectively (Table 3). Ghaffari et al. [13] also showed an increase in TFC and TPC in *Perovskia abrotanoides* as a result of being affected by blue light. Hinds et al. [22] revealed an increase in the polyphenolic content of black peppercorns in all the studied lights. Flavonoids and other phenolic compounds can increase in response to stress, and in the present research, blue-red and blue light were more effective in increasing such components in both studied species. Based on a review report, the response of plants can be different for the accumulation of polyphenolic compounds according to species, organ, phenological stage, and light intensities [23]. Darko et al. [24] highlighted the role of blue and red light in activating the xanthophyll cycle and inducing stress in plants. Liu et al. [25] also revealed that blue light can greatly increase the number of phenolic compounds in plants. In particular, it was suggested that blue light mainly increases the content of simple phenols, such as chlorogenic, gallic acid, and flavonoids [26].

### 2.4. DPPH Scavenging Activity

The highest antioxidant activity (lowest EC_50_) was observed in the DPPH scavenging assay in oregano for white LED light, while for marjoram, it was obtained under blue-red illumination (Table 3). Tohidi et al. [11] showed different reactions of species to antioxidant activity. In some cases, blue light increased the antioxidant activity of some plants such as Plectranthus amboinicus [27]. In our study, the higher total phenolic content was not directly correlated with antiradical activity, suggesting a selective increase in stronger antioxidants under specific conditions.

### 2.5. Inhibition of β-Carotene Bleaching

A similar trend as the one in the DPPH assay results was also observed in this model (Figure 2). The presence of different antioxidants can hinder β-carotene bleaching by quenching free radicals in the reaction mixture [28]. *O. vulgare* in the control possessed a high inhibition of β-carotene (79.3%). In this method, all extracts showed a greater ability to prevent oxidation of fatty acids than BHT at the same concentration (Figure 2).

LEDs can be a stressful condition for plants, so in these conditions, reactive oxygen species (ROS) can accumulate in plant organs. Consequently, metabolic activity will be elevated to protect plants against moderate photooxidative stress induced by LEDs [10,29]. Based on the previous literature, there are different trends regarding the use of LEDs on antioxidant activity of plants. For example, in lemon balm, blue-red LEDs revealed the highest effect on increasing ROS scavenging [10], while red light showed an increasing effect on DPPH removal activity in lettuce [30]. In pea sprouts, blue light led to a significant increase in antioxidant activity [25].

### 2.6. Physiological Traits

Chlorophyll a, chlorophyll b, and carotenoids concentrations were elevated under LED lights as compared with a control condition. Additionally, the activity of antioxidant enzymes, including catalase, guaiacol peroxidase, ascorbate peroxidase, as well as proline increased their concentration (Table 4).

The highest amount of chlorophyll a, chlorophyll b, and carotenoids was calculated in blue light for both species. The treatments of OM-B (0.78 mg g^−1^) and OV-V (0.71 mg g^−1^) displayed the highest amount in chlorophyll a, respectively. The highest (0.16 mg g^−1^) and lowest (0.08 mg g^−1^) amount in chlorophyll b were recorded in the OV-B and OM-R treatments, respectively (Table 4). The evaluation of antioxidant enzymes activity revealed that the highest amount of catalase was reported in OM-B (0.68 units mg^−1^ protein) and the lowest in OV-C (0.13 units mg^−1^ protein) (Table 4). In addition, the highest and lowest amount of guaiacol peroxidase were recorded in OV-B and OM-R, respectively. The highest amount of proline was reported in OV-B (0.65 μmolg^−1^), while the lowest was in OV-C (0.15 μmolg^−1^) (Table 4). OV-BR (5.13 units mg^−1^ protein) and OM-C (1.17 units mg^−1^ protein) demonstrated the highest and lowest amount of ascorbate peroxidase, respectively (Table 4).

Based on previously published literature, antioxidant enzymes increase for the purpose of defending against stress conditions that might be related to different factors like reactive oxygen species (ROS) [13]. In the present research, some light wavelengths such as the blue one provides a stress condition for plants that leads to the accumulation of more antioxidant enzymes. Another interpretation of such results is the high energy of a lower wavelength might lead to the increase of stress [13]. A similar trend is also obvious for other pigments such as carotenoids as well as stress related amino acids such as proline.

### 2.7. Multivariate Analysis

Species, treatment, and components were classified according to the principal component analysis (PCA) (Figure 3). Accordingly, three groups were obtained. Group 1 consisted of OV (C), OV (W), OM (C), and OM (W); Group 2 included OV (B), OV (R), and OV (BR); whereas OM (B), OM (BR), and OM (R) were classified into the third group. Group 1 included high essential oil samples, while Group 2 possesses a high thymol, carvacrol, and TPC content. Group 3 was marjoram with a high content of linalool, sabinene, terpinen-4-ol, and high TFC content.

Cluster analysis was also used to classify species and treatments based on the Ward method. This method of analysis also confirms the division obtained in PCA in most cases (Figure 4). According to clustering, two major groups were obtained. Group 1 included all control and white lights in both species. Group 2 consisted of the other colors that were included in both species. Therefore, it might be concluded that red, blue, and blue-red lights showed different trends for the accumulation of secondary metabolites in the two studied species.

## 3. Materials and Methods

### 3.1. Plant Materials and Cultivation

Oregano (*O. vulgare*) and marjoram (*O. majorana*) seeds were obtained through the medicinal plant collections of the Isfahan University of Technology. Oregano has previously been collected from Tabriz, East Azerbaijan, and Iran, and identified by Dr. Rahimmalek using Flora Iranica [31]. The voucher specimens were deposited in the Herbarium of Isfahan University of Technology with the code 15225. Marjoram has also been purchased from Pakan Bazr, Company, Isfahan, determined by Dr. Rahimmalek, and deposited in the herbarium with the code 15323.

The seeds were cultivated in 10 kg pots with a mixture of sand and silt loam soil (1:2 *v*/*v*). Seeds were sown in greenhouse conditions at 25 °C. The aerial parts were harvested just before flowering and dried in shade at room temperature.

### 3.2. LED Treatments

The LED chambers were set up according to the Sabzalian et al.’s [14] procedure. LED sheets with 120 1 W LEDs (0.25 A input current, OSRAM, Germany) were used to provide blue (460–475 nm), white (380–760 nm), red (650–665 nm), and blue-red (70%:30%) light. The temperature was set at 25 °C and the humidity at 60%. The voltage of the arrays was set to reach 300 μmol (photon) m^−2^ s^−1^ Control plants were kept at greenhouse conditions under ambient exposure to natural light.

### 3.3. Essential Oil Extraction

The dried leaves were applied for oil extraction, so that the leaves of plants were shade dried at room temperature (20–25 °C) over a period of seven days. For this purpose, 60 g of dried parts were used. Distillation was performed using the Clevenger-type apparatus. Then, 500 mL of distilled water was added and the samples were heated for 5 h. Finally, the oils were collected in a glass container. The oil content was calculated based on the dry matter of each sample in three replicates.

### 3.4. GC-MS Analysis

An Agilent 7890 gas chromatograph (Agilent Technologies, Palo Alto, CA, USA), with an HP-5MS capillary column (30 m × 0.25 mm, 0.25 μm film thickness) was applied for oil analysis. The conditions were as follows: the initial oven temperature was 60 °C for 4 min and, for the next steps, 260 °C was applied. The injector and detector temperatures were 290 °C and 300 °C, respectively. The carrier gas was helium with a flow rate of 2 mL/min. The gas chromatograph was coupled to an Agilent 5975 C (Agilent Technologies). An ionization voltage of 70 eV and an ion source temperature of 200 °C were applied for the mass spectrometry system.

Compounds were determined in an HP-5MS column using retention indices (RI) of C5–C24 n-alkanes and comparison with those reported in the literature using NIST 08 (National Institute of Standards and Technology, Gaithersburg, MD, USA) and Willey (Chem Station data system, Santa Clara, CA, United States). The Adams (2001) library was also used for determining the components.

### 3.5. Total Phenolic and Flavonoid Content

Total phenolic content (TPC) presented in the plants was evaluated using the Folin–Ciocalteu method as described by Pinelo et al. [32]. In brief, 6 g of dried plant material was extracted with 100 mL of 80% MeOH (24 h, shaken 60 r.p.m.). After centrifugation and filtration, the solution was submitted for further analyses.

The aluminum chloride method was applied to determine the total flavonoids content using quercetin as a standard [33]. The absorbance of the final mixture was measured at 510 nm. TFC was expressed as milligrams of quercetin equivalents per gram of the sample.

### 3.6. DPPH Scavenging Activity

The procedure of measuring the DPPH radical scavenging activity was used for this assay as previously described in [34,35,36]. For this, the extracts were prepared in the different concentrations of 50, 300, and 500 mg/L in methanol. The absorbance was reported at 517 nm against a blank (plant extract without DPPH) and BHT was applied as standard control for comparison. Finally, by formulating the extract concentrations against the scavenging activity, concentration of the samples was calculated based on EC50.

### 3.7. Antioxidant Activity Using β-Carotene-Linoleic Acid Model System of β-Carotene-Linoleic Acid

The antioxidant capacity of plant extracts was identified using β-carotene bleaching test [37]. To start, 10 mg of β-carotene (type I synthetic, Sigma–Aldrich, St. Louis, MO, USA) was dissolved in 10 mL of chloroform. The carotene–chloroform solution (0.2 mL) was added to a boiling flask containing 20 mg of linoleic acid and 200 mg Tween. Chloroform was removed using a rotary evaporator (Buchi 461, Uster, Switzerland) at 40 °C for 5 min and, to the residue, 50 mL of distilled water was added slowly, with vigorous agitation, to form an emulsion. Then, 5 ml of the emulsion was collected in one tube containing 0.2 mL of extracts prepared according to [38] and the absorbance was calculated at 470 nm against a blank, consisting of an emulsion without β-carotene. The tubes were placed in a 50 °C water bath and the oxidation of the emulsion was evaluated spectrophotometrically at 470 nm. The control samples included 10 mL of water instead of essential oils. Butylated hydroxytoluene (BHT; Sigma–Aldrich), a stable antioxidant, was applied as a synthetic antioxidant. The antioxidant capacity was stated as an inhibition percentage with reference to the control after 60 min of incubation.

### 3.8. Measurement of Chlorophyll and Carotenoids Concentrations

The amount of chlorophyll and carotenoids was determined according to the method developed by Lichtenthaler and Wellburn [39]. For this purpose, 0.2 g of fresh leaves were gently mixed with 10 mL of 80% (*v*/*v*) acetone. The solution was centrifuged (5810R, Eppendorf Refrigerated Centrifuge, Germany) for 10 min at 5000 rpm. Then, the absorbance of the produced extract was recorded at 645, 663, and 470 nm, respectively, using a spectrophotometer (U-1800 UV/VIS, Hitachi, Japan). The concentrations of carotenoids, chlorophyll a, and chlorophyll b were expressed as mg per g of leaf fresh weight.

### 3.9. Antioxidant Enzyme Activity Assays

Each of the LED Treatment samples (0.1 g) was ground in a mortar and pestle. Then, the powder was homogenized with 1 mL of Na-phosphate buffer (50 mM, pH 7) including 2 mM EDTA, 50 mM Tris−HCl, 2 mM α-dithiothreitol (DTT), 0.2% triton X-100, and 2% polyvinyl pyrrolidone (PVP). The homogenate solution was centrifuged at 15,000× *g* for 30 min at 4 °C. All stages of the extraction were performed at 4 °C. The supernatant of this extraction solution was used to measure the activities of the following antioxidant enzymes.

#### 3.9.1. GPOX (EC 1.11.1.7)

Guaiacol peroxidase (EC 1.11.1.7) activity was assessed using the method from Askari and Ehsanzadeh [40]. For this purpose, 3 mL extraction buffer, 3.35 μL guaiacol, and 4.51 μL H_2_O_2_ (30%) were added to the 50 μL extraction solution to measure guaiacol peroxidase activity. Then, the absorption changes were obtained at 240 nm for 2 min. Guaiacol peroxidase activity was reported as a unit per mg of protein.

#### 3.9.2. APX (EC 1.11.1.11)

Ascorbate peroxidase activity was evaluated with the method described by Nakano and Asada [41]. Briefly, 3 mL of extraction buffer was mixed with 4.51 μL H_2_O_2_ (30%), 0.1 mM EDTA, 100 μL of 5 mM ascorbate acid, and 50 μL extract. The decrease in 290 nm was recorded. Ascorbate peroxidase (APX) activity was expressed as unit per mg of protein.

#### 3.9.3. CAT (EC 1.11.1.6)

Examination of catalase activity (EC 1.11.1.6) was performed according to Chance and Maehly’s [42] method. Firstly, 50 μL extraction solution was mixed with 4.51 μL H_2_O_2_ (30%) and 3 mL extraction buffer (50 mM potassium phosphate buffer, pH 7.0 at 25 °C). The absorbance reducing data were read at 240 nm for 2 min. Catalase activity was calculated as a unit per mg of protein.

### 3.10. Proline

The method for proline evaluation was described by Bates et al. [43]. In the method, 0.2 g of the fresh mature studied leaves were homogenized in 10 mL of 3% sulphosalicylic acid and then the obtained solution was filtered. Next, 2 mL of each solution were mixed with 2 mL glacial acetic acid and 2 mL of ninhydrin reagent. The reaction mixtures were kept for an hour in a water bath and were instantly transferred to the ice bath. After adding 4 mL toluene to each sample and 30 min incubation, their absorbance was recorded at 520 nm. Finally, proline concentration was measured according to the standard curve that was obtained through specified concentrations of this amino acid.

### 3.11. Statistical Analyses

All the measurements of all the samples were carried out in triplicates. The analysis of variance was carried out using SAS ver. 9.2. LSD test (*p* < 0.05) to compare treatment means. Multivariate analyses such as principal component analysis (PCA) were also applied for biplot analysis of the components.

## 4. Conclusions

Today, improving the content and quality of essential oil in medicinal plants is of great importance in the food and pharmaceutical industries as it is related to the change in the nutritional, medicinal, and antioxidant properties of the final products. In addition, there is a growing interest in using greenhouses and indoor facilities instead of the field to grow medicinal species. Under these conditions, LEDs can be used as supplementary light or as the sole source of light to increase photosynthesis and metabolite accumulation. In the present investigation, essential oil content and chemical composition, as well as total polyphenols and antioxidant capacity of oregano and marjoram were studied in the context of their response to different spectra of light-emitting diode illumination. For essential oil content, the two species revealed different trends. However, in oregano, LED illumination caused a dramatic increase in thymol and cymene at the expense of linalool and terpinen-4-ol, which are the main constituents of plants under natural light. Conversely, in marjoram, the contribution of linalool and terpinen-4-ol decreased in favor of thymol and γ-terpinene only under red LEDs, while the composition under other light regimes was less affected. In the present study, blue-red LED light treatments produced the highest amount of the major monoterpene components with high antioxidant capacity in marjoram. So, for marjoram, this might suggest that its antioxidant capacity under light treatment can be related to the essential oil components, especially monoterpenes. However, in oregano, its antioxidant capacity has not been consistently reflected. Regarding nonvolatile antioxidants such as flavonoids, blue-red light was an efficient stimulant of their accumulation, also causing a significant increase in antioxidant potential only in marjoram. Therefore, the results of this research can provide new insights into influencing the essential oil profile of oregano and marjoram as important herb and spice plants. Finally, the results can help growers manipulate the composition and yield of beneficial antioxidant compounds.

## Figures and Tables

**Figure 1 molecules-28-03714-f001:**
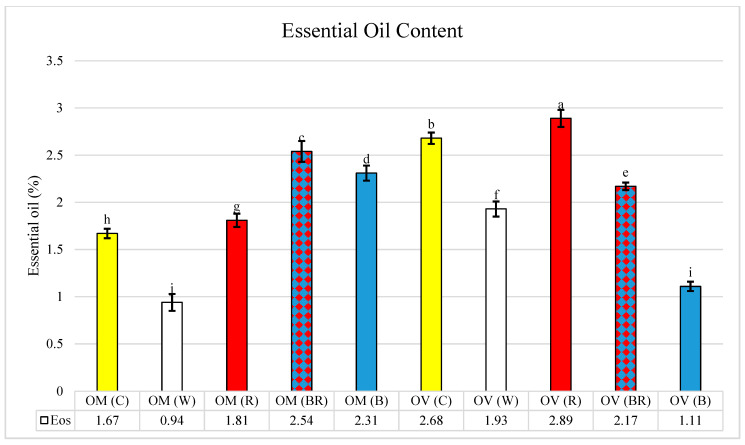
Variation of essential oil content in *Origanum majorana* (OM) and O. vulgare (OV) species grown under different light spectra: C—ambient natural light, B—blue LED, R—red LED, BR—7:3 mix of blue and red LED, W—white LED. The letters on the graph illustrates the mean differences of studied samples in different LED treatments.

**Figure 2 molecules-28-03714-f002:**
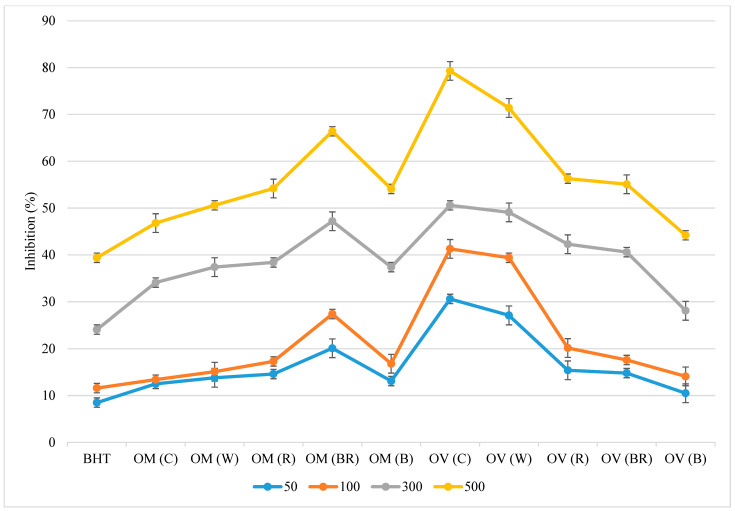
Inhibition of β-carotene bleaching system of Origanum majorana (OM) and O. vulgare (OV) as compared with BHT. The concentration steps were 50, 100, 300, and 500 ppm. C—ambient natural light, B—blue LED, R—red LED, BR—7:3 mix of blue and red LED, W—white LED.

**Figure 3 molecules-28-03714-f003:**
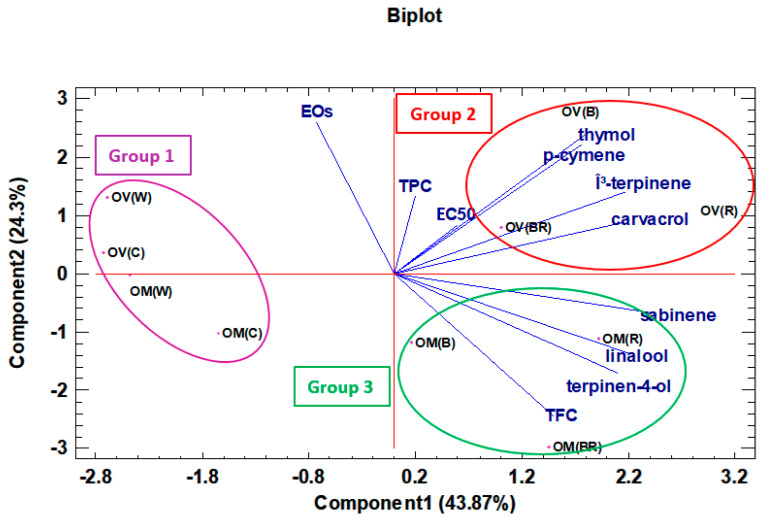
Principal component analysis (PCA) of *Origanum majorana* (OM) and *O. vulgare* (OV) species grown under different light spectra: C-ambient natural light, B-blue LED, R-red LED, BR-7:3 mix of blue and red LED, W-white LED.

**Figure 4 molecules-28-03714-f004:**
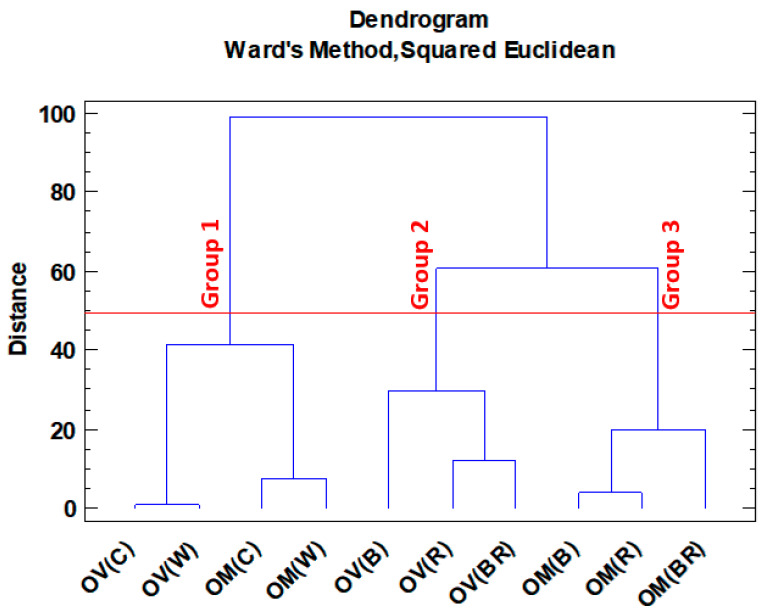
*Origanum majorana* (OM) and *O. vulgare* (OV) species grown under different light spectra: C—ambient natural light, B—blue LED, R—red LED, BR—7:3 mix of blue and red LED, W—white LED.

**Table 1 molecules-28-03714-t001:** Analysis of essential oils, total phenolics, total flavonoids, and antioxidant activity in *Origanum majorana* and *O. vulgare* under LED treatments.

Source of Variation	DF	Mean Squares				
		Essential Oil	Total Phenolics	Total Flavonoids	EC_50_	β-Carotene-Linoleic Acid
LED	4	3.62 **	848.10 **	10.33 **	143.564 **	15.426 **
Rep (LED)	10	0.023 *	0.084 **	0.056 *	979 *	148 *
Species	1	29.80 **	125.50 **	8.26 **	150.070 **	12.736 **
Species x LED	4	4.20 **	288.76 **	2.18 **	287.053 **	16.478 **
Error	10	0.005	0.013	0.007	28.62	24.51

DF: degree of freedom; *: significant at *p* ≤ 0.05; **: significant at *p* ≤ 0.01.

**Table 2 molecules-28-03714-t002:** Proportion of volatile compounds (%) of essential oils of Origanum majorana (OM) and O. vulgare (OV) species grown under different light spectra: C—ambient natural light, B—blue LED, R—red LED, BR—7:3 mix of blue and red LED, W—white LED. RI_exp_—experimental retention index; RI_lit_—literature (NIST17) retention index. Bolded are major compounds.

Composition (%)	RI_exp_	RI_lit._	OV-C	OV-B	OV-R	OV-BR	OV-W	OM-C	OM-B	OM-R	OM-BR	OM-W	Identification
α-thujene	928	929	0.6 ± 0.08	1.31 ± 0.02	0.84 ± 0.01	0.90 ± 0.01	0.48 ± 0.03	0.56 ± 0.04	0.13 ± 0.01	0.39 ± 0.02	2.09 ± 0.01	1.69 ± 0.03	IS, KI, MS
α-pinene	939	937	0.61 ± 0.01	2.02 ± 0.03	0.88 ± 0.02	0.98 ± 0.01	0.56 ± 0.02	0.68 ± 0.01	0.31 ± 0.03	0.46 ± 0.01	2.28 ± 0.02	0.38 ± 0.01	IS, KI, MS
**sabinene**	975	974	**6.17 ± 0.05**	**5.62 ± 0.04**	**3.42 ± 0.03**	**3.61 ± 0.03**	**3.98 ± 0.02**	**8.00 ± 0.05**	**4.82 ± 0.02**	**4.81 ± 0.06**	**4.56 ± 0.08**	**6.05 ± 0.07**	IS, KI, MS
β-pinene	976	979	1.37 ± 0.01	0.29 ± 0.01	0.69 ± 0.01	0.46 ± 0.01	0.78 ± 0.02	0.58 ± 0.01	1.32 ± 0.01	1.07 ± 0.02	0.78 ± 0.01	0.47 ± 0.02	IS, KI, MS
β-myrcene	992	991	0.43 ± 0.01	0.92 ± 0.02	1.47 ± 0.03	1.67 ± 0.07	1.05 ± 0.01	3.27 ± 0.01	2.17 ± 0.04	1.96 ± 0.01	1.14 ± 0.09	1.60 ± 0.01	IS, KI, MS
**α-terpinene**	1018	1017	**4.6 ± 0.02**	**2.16 ± 0.01**	**1.59 ± 0.01**	**1.29 ± 0.02**	**2.10 ± 0.01**	**3.75 ± 0.03**	**6.87 ± 0.06**	**7.63 ± 0.03**	**1.41 ± 0.01**	**9.52 ± 0.06**	IS, KI, MS
β-phellandrene	1026	1029	0.34 ± 0.01	0.2 ± 0.01	0.29 ± 0.06	0.32 ± 0.01	0.08 ± 0.02	0.93 ± 0.01	0.41 ± 0.01	0.30 ± 0.01	0.25 ± 0.03	0.83 ± 0.01	IS, KI, MS
** *p* ** **-cymene**	1026	1024	**7.39 ± 0.02**	**5.17 ± 0.03**	**18.24 ± 0.02**	**27.54 ± 0.01**	**25.48 ± 0.2**	**12.88 ± 0.1**	**4.64 ± 0.05**	**4.84 ± 0.02**	**11.59 ± 0.06**	**6.44 ± 0.04**	IS, KI, MS
limonene	1029	1030	1.53 ± 0.01	0.61 ± 0.01	1.20 ± 0.01	1.14 ± 0.02	0.26 ± 0.01	3.05 ± 0.01	1.19 ± 0.01	1.18 ± 0.04	0.30 ± 0.02	2.52 ± 0.03	IS, KI, MS
**γ-terpinene**	1063	1060	8.61 ± 0.02	6.75 ± 0.01	**12.70 ± 0.4**	**8.09 ± 0.5**	**9.25 ± 0.07**	**2.12 ± 0.01**	**3.30 ± 0.06**	**4.51 ± 0.04**	**9.37 ± 0.05**	**2.41 ± 0.06**	IS, KI, MS
fenchone	1082	1086	2.52 ± 0.01	2.93 ± 0.03	1.09 ± 0.03	1.86 ± 0.01	1.35 ± 0.01	2.00 ± 0.01	4.40 ± 0.03	3.10 ± 0.01	3.71 ± 0.01	1.04 ± 0.02	IS, KI, MS
terpinolene	1085	1088	1.52 ± 0.01	1.85 ± 0.01	0.96 ± 0.01	0.97 ± 0.03	0.89 ± 0.01	0.86 ± 0.01	3.25 ± 0.01	2.44 ± 0.01	0.91 ± 0.01	1.62 ± 0.01	IS, KI, MS
**linalool**	1097	1099	**20.11 ± 0.2**	**0.38 ± 0.01**	**12.04 ± 0.2**	**5.64 ± 0.1**	**10.69 ± 0.3**	**22.42 ± 0.2**	**18.82 ± 0.2**	**19.48 ± 0.4**	**7.92 ± 0.3**	**13.75 ± 0.2**	IS, KI, MS
fenchol	1110	1113	0.89 ± 0.01	0.74 ± 0.01	0.63 ± 0.01	0.59 ± 0.01	0.58 ± 0.01	0.61 ± 0.01	1.34 ± 0.01	1.16 ± 0.01	0.51 ± 0.01	1.64 ± 0.04	IS, KI, MS
α-campholenal	1123	1125	0.92 ± 0.02	0.85 ± 0.01	0.58 ± 0.01	1.03 ± 0.01	0.93 ± 0.01	1.82 ± 0.02	1.22 ± 0.01	0.94 ± 0.01	1.75 ± 0.01	0.59 ± 0.01	KI, MS
**terpinen-4-ol**	1178	1182	**19.02 ± 0.1**	**1.93 ± 0.08**	**7.40 ± 0.07**	**3.81 ± 0.02**	**6.95 ± 0.08**	**8.70 ± 0.07**	**11.42 ± 0.09**	**13.39 ± 0.1**	**2.83 ± 0.04**	**12.11 ± 0.09**	IS, KI, MS
α-terpineol	1192	1189	2.67 ± 0.01	2.93 ± 0.02	1.20 ± 0.03	2.54 ± 0.01	2.18 ± 0.03	2.40 ± 0.01	1.03 ± 0.01	2.21 ± 0.06	3.53 ± 0.02	5.94 ± 0.01	IS, KI, MS
geraniol	1256	1255	0.2 ± 0.01	1.1 ± 0.01	1.09 ± 0.01	1.12 ± 0.01	1.13 ± 0.01	0.42 ± 0.01	2.17 ± 0.01	1.54 ± 0.01	0.38 ± 0.01	1.76 ± 0.01	IS, KI, MS
linalyl acetate	1254	1257	1.21 ± 0.01	2.59 ± 0.01	1.34 ± 0.01	1.52 ± 0.03	1.50 ± 0.01	1.24 ± 0.03	2.47 ± 0.05	1.73 ± 0.01	2.03 ± 0.04	2.21 ± 0.04	IS, KI, MS
bornyl acetate	1279	1285	0.32 ± 0.01	1.12 ± 0.01	0.47 ± 0.01	0.35 ± 0.01	0.22 ± 0.02	0.63 ± 0.01	1.66 ± 0.01	1.31 ± 0.01	0.38 ± 0.01	1.06 ± 0.01	IS, KI, MS
**thymol**	1293	1291	**11.91 ± 0.2**	**48.26 ± 0.5**	**19.99 ± 0.4**	**21.64 ± 0.3**	**23.19 ± 0.6**	**18.48 ± 0.6**	**20.38 ± 0.7**	**17.47 ± 0.3**	**35.06 ± 0.7**	**20.16 ± 0.5**	IS, KI, MS
**carvacrol**	1300	1299	**3.74 ± 0.1**	**4.37 ± 0.09**	**8.33 ± 0.07**	**9.13 ± 0.08**	**3.56 ± 0.05**	**0.70 ± 0.03**	**5.07 ± 0.06**	**5.23 ± 0.07**	**3.23 ± 0.06**	**2.63 ± 0.05**	IS, KI, MS
β-caryophyllene	1419	1419	1.75 ± 0.03	2.81 ± 0.01	1.43 ± 0.01	1.41 ± 0.01	0.51 ± 0.01	3.03 ± 0.01	1.32 ± 0.03	1.31 ± 0.01	0.80 ± 0.01	2.16 ± 0.03	IS, KI, MS
aromadendrene	1438	1440	0.2 ± 0.01	0.5 ± 0.01	0.25 ± 0.02	0.08 ± 0.01	0.20 ± 0.02	0.25 ± 0.01	0.31 ± 0.01	0.19 ± 0.02	0.51 ± 0.05	0.38 ± 0.01	KI, MS
α-humulene	1451	1454	0.42 ± 0.01	1.21 ± 0.02	0.58 ± 0.01	0.90 ± 0.02	0.57 ± 0.01	0.32 ± 0.03	0.59 ± 0.03	0.40 ± 0.01	2.09 ± 0.02	0.97 ± 0.01	IS, KI, MS

**Table 3 molecules-28-03714-t003:** Effects of LED illumination on total phenolics, total flavonoids, and DPPH scavenging activity in *Origanum majorana* and *O. vulgare*.

LED	Species	Total Phenolics (mg TAE/g DW)	Total Flavonoids (mg QUE/g DW)	EC_50_ (μg/mL)
Control	*O. majorana*	34.12 ± 0.04 ^g^	3.01 ± 0.01 ^e^	1032.9 ± 0.1 ^b^
	*O. vulgare*	31.74 ± 0.02 ^h^	1.89 ± 0.01 ^g^	371.80 ± 0.4 ^j^
White	*O. majorana*	65.18 ± 0.01 ^a^	1.92 ± 0.02 ^g^	975.73 ± 0.5 ^c^
	*O. vulgare*	39.35 ± 0.03 ^f^	1.73 ± 0.01 ^h^	455.6 ± 0.6 ^i^
Blue	*O. majorana*	53.17 ± 0.05 ^c^	4.11 ± 0.01 ^b^	886.77 ± 0.2 ^d^
	*O. vulgare*	61.26 ± 0.01 ^b^	2.16 ± 0.03 ^f^	1160.3 ± 0.1 ^a^
RED	*O. majorana*	51.05 ± 0.07 ^d^	3.29 ± 0.02 ^d^	782.03 ± 0.1 ^e^
	*O. vulgare*	43.28 ± 0.04 ^e^	3.76 ± 0.01 ^c^	704.73 ± 0.3 ^g^
Blue-Red	*O. majorana*	26.50 ± 0.01 ^i^	6.51 ± 0.02 ^a^	477.3 ± 0.6 ^h^
	*O. vulgare*	33.94 ± 0.08 ^g^	4.06 ± 0.01 ^b^	755.0 ± 0.2 ^f^

Means in each column, means followed by same letter are not significantly different according to LSD’s test at 5% level.

**Table 4 molecules-28-03714-t004:** Effects of LED illumination on traits on *Origanum majorana* (OM) and *O. vulgare* (OV) species grown under different light spectra.

Proline (μmol g^−1^)	Catalase (Units mg^−1^ Protein)	Ascorbate Peroxidase (Units mg^−1^ Protein)	Guaiacol Peroxidase (Units mg^−1^ Protein)	Carotenoids Concentration (mg g^−1^)	Chlorophyll b Concentration (mg g^−1^)	Chlorophyll a Concentration (mg g^−1^)	Treatments
0.15 ± 0.02	0.13 ± 0.01	2.14 ± 0.01	0.11 ± 0.04	0.12 ± 0.01	0.08 ± 0.03	0.23 ± 0.05	OV-C
0.65 ± 0.01	0.61 ± 0.03	3.24 ± 0.02	1.01 ± 0.01	0.22 ± 0.04	0.16 ± 0.01	0.71 ± 0.01	OV-B
0.25 ± 0.02	0.22 ± 0.01	5.01 ± 0.04	0.22 ± 0.06	0.11 ± 0.02	0.09 ± 0.06	0.32 ± 0.05	OV-R
0.56 ± 0.1	0.52 ± 0.01	5.13 ± 0.03	0.72 ± 0.01	0.13 ± 0.01	0.13 ± 0.04	0.62 ± 0.02	OV-BR
0.37 ± 0.04	0.35 ± 0.02	4.16 ± 0.01	0.43 ± 0.01	0.18 ± 0.04	0.11 ± 0.02	0.45 ± 0.01	OV-W
0.23 ± 0.03	0.19 ± 0.01	1.17 ± 0.01	0.19 ± 0.02	0.13 ± 0.1	0.07 ± 0.01	0.29 ± 0.03	OM-C
0.64 ± 0.01	0.68 ± 0.05	3.21 ± 0.02	0.91 ± 0.02	0.20 ± 0.06	0.14 ± 0.02	0.78 ± 0.02	OM-B
0.16 ± 0.02	0.14 ± 0.04	2.02 ± 0.06	0.12 ± 0.03	0.12 ± 0.1	0.08 ± 0.04	0.24 ± 0.01	OM-R
0.48 ± 0.01	0.49 ± 0.01	2.11 ± 0.1	0.63 ± 0.01	0.14 ± 0.05	0.12 ± 0.01	0.59 ± 0.01	OM-BR
0.21 ± 0.01	0.29 ± 0.02	1.94 ± 0.03	0.29 ± 0.02	0.16 ± 0.02	0.10 ± 0.05	0.39 ± 0.04	OM-W

C: ambient natural light, B: blue LED, R: red LED, BR: 7:3 mix of blue and red LED, W: white LED.

## Data Availability

The data will be available based on request.

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
