# Peer review of "Essential Oil Composition and Antioxidant Activity of Oregano and Marjoram as Affected by Different Light-Emitting Diodes"

_molecules, 2023, doi:10.3390/molecules28093714_

Round 1

Reviewer 1 Report

Please arrange the title according to Molecules requirements: LED Illumination Affects Essential Oil Composition, Antioxidant Activity and Antioxidative Enzymes in Oregano (Origanum vulgare L.) and Marjoram (Origanum majorana L.)

4 Department of Pharmaceutical Biology and Biotechnology, Botanical Garden of Medicinal Plants, Wroclaw Medical University, 50-367 Poland E-mail?

Lines 24 and 25 have an inaccurate sentence. Maybe: In oregano, the thymol content ranged from 11.91% in control to 48.26% in blue light illumination, while in marjoram it varied from 17.47% in red to 35.06% in blue-red light illumination.

In some places it is written ml (e.g., in lines 98, 106), in others mL. It is necessary to unify.

Why is the total phenolic content expressed as tannic acid equivalent and not gallic acid, as most researchers do, or the dominant rosmarinic acid [Oniga I, Pușcaș C, Silaghi-Dumitrescu R, Olah NK, Sevastre B, Marica R, Marcus I, Sevastre-Berghian AC, Benedec D, Pop CE, Hanganu D. Origanum vulgare ssp. vulgare: Chemical Composition and Biological Studies. Molecules. 2018 Aug 19;23(8):2077. doi: 10.3390/molecules23082077.]?

Line 97: Not sure what parts, what drying method and parameters, was it milled?

Line 98:  Incorrect: The extraction was performed using the Clevenger type.

Line 115: Total phenolic content (TPC) was evaluated using the Folin–Ciocalteu method as described by Gharibi et al. [16]. But Gharibi seems to be one of the authors of this article and cites his article without justification because it does not describe the methodology and instead cites another source. Thus, a more accurate methodology should be specified.

Where is Fig. S1?

In Figure 1, what is Eos?

Table 3. What is majorana and vulgare? It is Origanum majorana and Origanum vulgare?

Table 3. What is a, b, c, et al.? It is Duncan‘s Multiple range Test?  In 2.11. Statistical analysis you wrote that you use the LSD test...

What in Table 2 is degree of freedom; *: significant at P≤0.05; **: significant at P≤0.01;?

Discussion section is missing.

Author Response

Response to reviewers’ comments

Dear Dr. Kane Tang

Editor

Molecules,

Manuscript title: LED Illumination Affects Essential Oil Composition, Antioxidant Activity and Antioxidative Enzymes in Oregano (Origanum vulgare L.) and Marjoram (Origanum majorana L.)

Manuscript ID:

The authors would like to thank the editor and the reviewers for their helpful and well-reasoned comments and suggestions. We have carefully considered their comments in preparing our revision. We are now submitting a revised manuscript, with the changes marked in Blue. Below, we also provide a point-by-point response explaining how we have addressed each of the reviewer's comments:

Reviewer #1:

C1: Please arrange the title according to Molecules requirements: LED Illumination Affects Essential Oil Composition, Antioxidant Activity and Antioxidative Enzymes in Oregano (Origanum vulgare L.) and Marjoram (Origanum majorana L.)

According to Reviewer’s suggestion title was changed into:

 Essential Oil Composition and Antioxidant Activity of Oregano and Marjoram as Affected by Different Light-Emitting Diodes

C2: 4 Department of Pharmaceutical Biology and Biotechnology, Botanical Garden of Medicinal Plants, Wroclaw Medical University, 50-367 Poland E-mail?

Answer: Thanks, fixed.

C3: Lines 24 and 25 have an inaccurate sentence. Maybe: In oregano, the thymol content ranged from 11.91% in control to 48.26% in blue light illumination, while in marjoram it varied from 17.47% in red to 35.06% in blue-red light illumination.

Answer: Thanks, it was corrected.

C4: In some places it is written ml (e.g., in lines 98, 106), in others mL. It is necessary to unify.

Answer: It was revised in the text.

C5: Why is the total phenolic content expressed as tannic acid equivalent and not gallic acid, as most researchers do, or the dominant rosmarinic acid [Oniga I, Pușcaș C, Silaghi-Dumitrescu R, Olah NK, Sevastre B, Marica R, Marcus I, Sevastre-Berghian AC, Benedec D, Pop CE, Hanganu D. Origanum vulgare ssp. vulgare: Chemical Composition and Biological Studies. Molecules. 2018 Aug 19;23(8):2077. doi: 10.3390/molecules23082077.]?

Answer:

Thank you for remark. We agree with it. From the other hand according to literature, both systems are used for expression of phenolic compounds. Due to our previous experience, we have decided to use tannic acid equivalent, see:

Rahimmalek, M., Afshari, M., Sarfaraz, D. and Miroliaei, M., 2020. Using HPLC and multivariate analyses to investigate variations in the polyphenolic compounds as well as antioxidant and antiglycative activities of some Lamiaceae species native to Iran. Industrial Crops and Products154, p.112640.

Tohidi, Behnaz, Mehdi Rahimmalek, and Ahmad Arzani. "Essential oil composition, total phenolic, flavonoid contents, and antioxidant activity of Thymus species collected from different regions of Iran." Food chemistry 220 (2017): 153-161.

Also, this system was used by other authors, e.g. Bizuayehu, D., Atlabachew, M. and Ali, M.T., 2016. Determination of some selected secondary metabolites and their invitro antioxidant activity in commercially available Ethiopian tea (Camellia sinensis). SpringerPlus5(1), pp.1-9.

C6: Line 97: Not sure what parts, what drying method and parameters, was it milled?

Answer: Thanks, it was corrected.

C7: Line 98:  Incorrect: The extraction was performed using the Clevenger type.

Answer: We changed word “extraction” into “distillation”, which is more accurate in case of EOs.

C8: Line 115: Total phenolic content (TPC) was evaluated using the Folin–Ciocalteu method as described by Gharibi et al. [16]. But Gharibi seems to be one of the authors of this article and cites his article without justification because it does not describe the methodology and instead cites another source. Thus, a more accurate methodology should be specified.

Answer: Thanks, fixed.

C9: Where is Fig. S1?

Answer: Thanks, it was corrected. It was deleted in a manuscript.

 In Figure 1, what is Eos?

Answer: the meaning of “EOs” was explained in the text. This is a common used abbreviation for essential oils. We explained it in beginning of introduction.

C10: Table 3. What is majorana and vulgare? It is Origanum majorana and Origanum vulgare?

Answer: Thanks, they were corrected.

C11: Table 3. What is a, b, c, et al.? It is Duncan‘s Multiple range Test?  In 2.11. Statistical analysis you wrote that you use the LSD test...

Right. Information was added that means in each column, means followed by same letter are not significantly different ac-cording to LSD’s test at 5% level. We added this information bellow the Table.

C12: What in Table 2 is degree of freedom; *: significant at P≤0.05; **: significant at P≤0.01;?

Reviewer is right. We deleted unnecessary information.

C13: Discussion section is missing.

According to journal requirements, we merged the results with discussion

Reviewer 2 Report

It will be necessary to correct the following topics:

(214) The authors say “In both species, red- and blue-red illumination resulted in a higher increase in oil content than the other light spectra” but in Figure 1 OV- C has essential oil percentage >2,5 % and OV-RB  <2,5 %.

(239)The authors say: “. An interesting point in the present investigation was the elevation of thymol and linalool in marjoram treated with red light (Table 2), but this table shows linalool values of 22.42±0.2 for OM-C and 19.48±0.4 for OM-R.

Author Response

Response to reviewers’ comments

Dear Dr. Kane Tang

Editor

Molecules,

Manuscript title: LED Illumination Affects Essential Oil Composition, Antioxidant Activity and Antioxidative Enzymes in Oregano (Origanum vulgare L.) and Marjoram (Origanum majorana L.)

Manuscript ID:

The authors would like to thank the editor and the reviewers for their helpful and well-reasoned comments and suggestions. We have carefully considered their comments in preparing our revision. We are now submitting a revised manuscript, with the changes marked in Blue. Below, we also provide a point-by-point response explaining how we have addressed each of the reviewer's comments:

Reviewer #2:

It will be necessary to correct the following topics:

C1: (214) The authors say “In both species, red- and blue-red illumination resulted in a higher increase in oil content than the other light spectra” but in Figure 1 OV- C has essential oil percentage >2,5 % and OV-RB  <2,5 %.

 Reviewer’s comment was true, we corrected sentence.

C2: (239)The authors say: “. An interesting point in the present investigation was the elevation of thymol and linalool in marjoram treated with red light (Table 2), but this table shows linalool values of 22.42±0.2 for OM-C and 19.48±0.4 for OM-R.

We agree, sentence was corrected.

We hope we have addressed all the comments and questions and hope the manuscript is acceptable for publication in Molecules.

Yours truly

Antoni Szumny and Shima Gharibi

In addition, we correct the manuscript in professional English editing service. The appropriate certificate was included.

Round 2

Reviewer 1 Report

Thanks to the authors for the correction.

Author Response

Response to reviewers’ comments

Dear Dr. Kane Tang

Editor

Molecules,

Manuscript title: LED Illumination Affects Essential Oil Composition, Antioxidant Activity and Antioxidative Enzymes in Oregano (Origanum vulgare L.) and Marjoram (Origanum majorana L.)

Manuscript ID:

The authors would like to thank the editor and the reviewers for their helpful and well-reasoned comments and suggestions. We have carefully considered their comments in preparing our revision. We are now submitting a revised manuscript, with the changes marked in green. Below, we also provide a point-by-point response explaining how we have addressed each of the reviewer's comments. Also, we precise that we analyzed total phenolics and flavonoids in plant extracts. Antioxidant activity of was measured for extracts, not EOs. It was our mistake in materials and methods section and other parts of manuscript. Appropriate corrections were done.

Lines 73-76: It is not clear if the phenolic and flavonoid contents were determined on the essential oil or an extract.

Information corrected. We added appropriate information about the fact, that phenolics and flavonoid content was measured in plant extract. Changes are marked in green.

Section 2.3: Only essential oil is described. Apparently no extract was obtained for the total phenolics and flavonoids assays.

Missed information was added into section 2.5.

Section 2.5: Were the total phenolics and flavonoids assays carried out on the essential oil? Please make this clear.We added necessary information (in green)

Section 2.8: An acetone extract was obtained for these assays.

Yes, as many other authors we analyzed the chlorophyll and carotenoids in acetone solutions. See e.g. Chazaux, M., Schiphorst, C., Lazzari, G., & Caffarri, S. (2022). Precise estimation of chlorophyll a, b and carotenoid content by deconvolution of the absorption spectrum and new simultaneous equations for Chl determination. The Plant Journal, 109(6), 1630-1648.

Table 1: Were these phytochemicals based on essential oils? Please make this clear in the title.

Title changed according to Editor's comment

Table 2: The highest phenolic contents (thymol+carvacrol) are for OV-B (around 53%) and for OM-BR (around 38%). However, these concentrations do not agree with the concentrations of total phenolics reported in Table 3 (highest phenolics in OM-W and lowest in OM-BR). This suggests there is something wrong in the total phenolics assay.

We agree that it could seem a little bit strange. But, we evaluated only the methanolic solution obtained from the plant. In such type of extraction, EOs are not totally dissolved in this mixture. We agree, that extraction with pure EtOH or with anhydrous MeOH could affect with  final mixture that contain both, i.e. extremely non polar EOs and polar phenolics and polyphenols.Moreover, the thymol+carvacrol is based on essential oil analysis and for total phenolic content we used methanolic extract that lead to obtain different results that might not related to the essential oil data.

Table 2 shows concentrations of total flavonoids in every sample. However, there are no flavonoids in the essential oils; this test for flavonoids gives incorrect information.We added information, that it was analyzed in extracts, not EOs.

Lines 297-299: "In our study, the higher total phenolic content was not directly correlated with antiradical activity, suggesting a selective increase in stronger antioxidants under specific conditions."

Which of the essential oil components are stronger radical scavengers than thymol or carvacrol?

Antioxidant activity was evaluated based on plant polar extracts, not EOs.
